# Survival and Chemosensitivity in Advanced High Grade Serous Epithelial Ovarian Cancer Patients with and without a BRCA Germline Mutation: More Evidence for Shifting the Paradigm towards Complete Surgical Cytoreduction

**DOI:** 10.3390/medicina58111611

**Published:** 2022-11-08

**Authors:** Diederick De Jong, Mohamed Otify, Inga Chen, David Jackson, Kelum Jayasinghe, David Nugent, Amudha Thangavelu, Georgios Theophilou, Alexandros Laios

**Affiliations:** 1ESGO Center of Excellence for Advanced Ovarian Cancer Surgery, Department of Gynaecological Oncology, St James’s University Hospital, Leeds Teaching Hospitals Trust, Leeds LS9 7TF, UK; 2Department of Obstetrics and Gynaecology, St James’s University Hospital, Leeds Teaching Hospitals Trust, Leeds LS9 7TF, UK; 3Department of Medical Oncology, St James’s University Hospital, Leeds Teaching Hospitals Trust, Leeds LS9 7TF, UK

**Keywords:** BRCA status, complete cytoreduction, chemotherapy response score, epithelial ovarian cancer, CA125

## Abstract

*Background and Objectives*: Approximately 10–15% of high-grade serous ovarian cancer (HGSOC) cases are related to BRCA germline mutations. Better survival rates and increased chemosensitivity are reported in patients with a BRCA 1/2 germline mutation. However, the FIGO stage and histopathological entity may have been confounding factors. This study aimed to compare chemotherapy response and survival between patients with and without a BRCA 1/2 germline mutation in advanced HGSOC receiving neoadjuvant chemotherapy (NACT). *Materials and Methods*: A cohort of BRCA-tested advanced HGSOC patients undergoing cytoreductive surgery following NACT was analyzed for chemotherapy response and survival. Neoadjuvant chemotherapy served as a vehicle to assess chemotherapy response on biochemical (CA125), histopathological (CRS), biological (dissemination), and surgical (residual disease) levels. Univariate and multivariate analyses for chemotherapy response and survival were utilized. *Results*: Thirty-nine out of 168 patients had a BRCA ½ germline mutation. No differences in histopathological chemotherapy response between the patients with and without a BRCA ½ germline mutation were observed. Survival in the groups of patients was comparable Irrespective of the BRCA status, CRS 2 and 3 (HR 7.496, 95% CI 2.523–22.27, *p* < 0.001 & HR 4.069, 95% CI 1.388–11.93, *p* = 0.011), and complete surgical cytoreduction (*p* = 0.017) were independent parameters for a favored overall survival. *Conclusions*: HGSOC patients with or without BRCA ½ germline mutations, who had cytoreductive surgery, showed comparable chemotherapy responses and subsequent survival. Irrespective of BRCA status, advanced-stage HGSOC patients have a superior prognosis with complete surgical cytoreduction and good histopathological response to chemotherapy.

## 1. Introduction

Ovarian cancer is the sixth most common cancer of women in the United Kingdom with about 7443 new cases diagnosed from 2015–2017. The majority of these patients are diagnosed in the advanced stages of disease (FIGO stage III–IV). The 1- and 5-year survival of these patients has been estimated as 54–73% and 13–27%, respectively [1,2]. The group of high grade epithelial ovarian related cancer (HGSOC) includes all high grade epithelial ovarian, tubal, and primary peritoneal cancers [3]. 

About 15% of HGSOC cases are related to germline mutations in BRCA1 or BRCA2 genes [4]. The BRCA germline mutation status has been reported to be a prognostic factor in women with ovarian cancer. A loss of BRCA function in women with ovarian cancer is associated with impaired tumor ability to perform double stranded DNA repair by homologous recombination. This may result in increased tumor sensitivity to platinum-based chemotherapy [5,6]. Women with HGSOC and germline BRCA mutations have been reported to have improved 5-year overall survival, compared to women without these mutations [6,7]. 

The standard therapy of patients with an advanced stage HGSOC consists of a combination of platinum-based chemotherapy and cytoreductive surgery [8]. Different treatment regimens have been developed using this approach. Neoadjuvant chemotherapy (NACT) followed by surgical cytoreduction and subsequent adjuvant chemotherapy has been associated with similar outcomes as the traditional approach of surgical cytoreduction followed by six cycles of chemotherapy [9,10]. Advanced ovarian cancer patients with BRCA1/2 germline mutation, treated with platinum-based neo adjuvant chemotherapy prior to surgery, may be expected to have less tumor volume at the time of surgical cytoreduction as compared to those patients without a BRCA1/2 germline mutation. 

We hypothesized that patients with BRCA1/2 germline mutations have better response to neoadjuvant chemotherapy (NACT) and that surgical cytoreduction in these patients may be less extended to achieve a complete surgical cytoreduction. In this study we analyzed the radiological, histopathological, and biochemical response to NACT as well as the surgical and survival outcomes in all patients with an advanced stage HGSOC whose BRCA germline status was tested. 

## 2. Materials and Methods

### 2.1. Selection of Patients, Data Collection, and Study Design 

For this study all patients with an advanced stage HGSOC who underwent surgical cytoreduction following NACT between October 2013 and October 2018 at a tertiary referral center for ovarian cancer surgery and were tested for a BRCA germline mutation were included in the analysis. Treatment and follow-up data was collected until December 2020. All patients had surgical cytoreduction by a certified and accredited Gynecologic Oncologist. Staging was defined by the 2014 International Federation of Gynaecology and Obstetrics (FIGO) staging system. [11] Excluded were those patients with a synchronous primary malignancy, those with recurrent ovarian malignancy and those who had surgery in the emergency setting (Figure 1). The follow-up was quarterly over the first two years, then biannually for a further two years then a final annual review when at five years patients could be discharged. Prospectively collected data from this cohort of women were retrieved from the hospital wide database “Patient Pathway Manager” PPM [12]. This study was approved by the Institutional Review Board (MO20/133163/18.06.20) and performed according to the standards outlined in the Declaration of Helsinki. 

### 2.2. Workup and Chemotherapy

In this analysis, the age was defined as age at the time of diagnosis. The performance status (PS) was determined at the initial presentation [13]. Serum CA125 levels were measured prior to the first course of NACT, and subsequently within two weeks prior to cytoreductive surgery. The CA125 response to chemotherapy, defined as ΔCA125, was calculated as the difference between these levels. BRCA counseling and testing was offered to patients according to NICE guidelines [14]. BRCA testing was carried out as published previously [15]. All the patients had pre-treatment physical examination, CT imaging of chest abdomen and pelvis, and histological diagnosis by assessment of either image-guided or laparoscopically obtained histological biopsy. The results of pre-treatment workup were discussed at our multi-disciplinary team (MDT) followed by further management recommendations. Chemotherapy in the neo-adjuvant setting consisted of three courses of carboplatin and paclitaxel (CPT) chemotherapy prior to surgery followed by three courses of CPT with or without bevacizumab post-surgery. The addition of bevacizumab was based on the presence of post-operative residual disease [16]. Single agent carboplatin (CP) instead of CPT was used in patients aged above 80 years and in those with a WHO PS > 2.

### 2.3. Surgical Procedure

The Surgical cytoreduction was performed by an abdominal midline incision with sampling of any ascitic fluid, total hysterectomy, bilateral salpingo-oophorectomy and omentectomy as the bare minimum. In an effort to achieve a complete surgical cytoreduction, the procedure could be extended with stripping of diaphragm and peritoneum, stripping of the mesentery, wedge resection of the liver, (partial) gastrectomy, cholecystectomy, splenectomy, pancreas tail resection, adrenalectomy, small and/or large bowel resection with or without stoma formation, appendicectomy, and lymphnode dissection.

### 2.4. Primary and Secondary Outcome Parameters

The primary outcome parameters were overall survival (OS), calculated from the date of diagnosis to the date of disease specific death or last follow-up. Secondary outcome parameters were progression-free survival (PFS), calculated from date of diagnosis to date of confirmed recurrence. Other secondary parameters included PS, CA125 levels, radiologic response to NACT, preoperative extent of disease, histologic response to chemotherapy, complexity of the surgical procedure, and residual tumor. 

The radiological response to NACT was categorized by use of RECIST criteria as published previously [17]. The preoperative extent of disease was grouped according to GOG classification [18]; the minimal disease (MD) group had tumor limited to the pelvis and retroperitoneal (nodal) metastasis. The abdominal peritoneal disease (APD) group had disease limited to the pelvis and abdomen but excluding the liver, spleen, gallbladder, pancreas, or diaphragm, with or without retroperitoneal spread. The upper abdominal disease (UAD) group had disease affecting the pelvis with or without lower abdominal and retroperitoneal disease, plus involvement of at least one of the following: liver, spleen, gallbladder, pancreas, or diaphragm. Histologic chemotherapy response score (CRS) in tissue obtained at surgery was scored as CRS 1 (no or minimal tumor response), CRS 2 (appreciable tumor response with residual tumor), or CRS 3 (complete or near-complete response) [19]. The intensity of the surgical procedure was scored according to the surgical complexity score (SCS) [20]. The biochemical response was measured by the ΔCA125, which was the difference between CA125 level before and after NACT. Residual disease (RD) was categorized according the size of the remaining tumor nodules at the end of the surgical procedure. To align with other surgical specialties we adopted the Sugarbaker classification for completeness of cytoreduction [21]. Complete cytoreduction of tumor was defined as a no measurable macroscopic RD (CC 0) or <2.5 mm RD (CC 1). Incomplete cytoreduction was defined as 2.5 mm < RD < 2.5 cm (CC 2) or RD ≥ 2.5 cm.

### 2.5. Statistical Analysis

Characteristics of the patients according to group were presented as means +/− SD or percentage. Differences between the groups of advanced HGSOC patients with and without a BRCA1/2 germline mutation was analyzed for normal distribution using D’Agostino-Pearson test; to test differences, for normally distributed data the Student T-test and for non-parametric data the Mann-Whitney test was used. The CA125 levels were log transformed before evaluation with the Kruskal–Wallis one-way analysis of variance. Categorical data was presented as frequency and per cent and compared using the Chi Square test or Fisher’s Exact test, as indicated. Survival was analyzed Kaplan-Meier and the Mantel-Cox log-rank test for comparison.

Univariate and multivariate analyses were performed using Ordered Logistic Regression. Independent variables found to have a *p* value < 0.1 in the univariate analysis were then combined in a multivariate analysis. All tests were two sided and *p* < 0.05 was considered statistically significant for all tests. The software package Stata/MP 13.0 (StataCorp, College Station, TX, USA) was employed for data analysis.

## 3. Results

### 3.1. Patient and Tumor Characteristics

Between 1 October 2013 and 1 October 2018, a cohort of 178 patients with FIGO stage III–IV HGSOC having surgical cytoreduction after 3–4 courses of neo-adjuvant chemotherapy and who were tested for a BRCA germline mutation was identified. Median follow up was 58 months. Excluded were patients who had surgery for recurrent disease (n = 3), those who had emergency surgery for bowel obstruction by the colorectal surgeons (n = 2), and those patients with a metastatic synchronous tumor (n = 5). Details of the study population are shown in Figure 1. A total of 129 patients had no BRCA germ-line mutation whereas 24 and 15 patients had a BRCA-1 and BRCA-2 germline mutation, respectively. Overall the mean age of the patients in our studied cohort was 64.4 ± 9.8 years. As expected, the patients with a BRCA 1/2 germline mutation were younger as compared to those who tested negative for a BRCA 1/2, 58.0 and 66.3 years respectively (*p* = 0.00001). Performance status, index CA 125 levels, FIGO stage, histology, primary site of disease, tumor differentiation, and NACT regimen were similar in the groups of patients with and without a BRCA 1/2 germline mutation. Baseline characteristics are displayed in Table 1. 

### 3.2. Response to Chemotherapy and Surgical Cytoreduction

The biochemical response to NACT, represented by median proportional fall in CA125 (ΔCA125), was 1111 (range 27–30715) U/mL for the patients with a BRCA 1/2 germline mutation versus 641 (range 23–17900) U/mL for those without a BRCA 1/2 germline mutation; *p* = 0.247, NS. The majority of patients with a BRCA 1/2 germline mutation and those without a BRCA 1/2 germline mutation had a partial or complete RECIST response following 3 courses of NACT, respectively 89.7 and 86.0%; *p* = 0.677, NS. There was no difference in histological response to NACT, represented by CRS, between HGSOC patients with and those without a BRCA 1/2 germline mutation. The removed surgical specimen showed a complete or near complete histopathological chemotherapy response score (CRS 3) in 10.3% of the patients with a BRCA 1/2 germline mutation versus 16.3% in those without, *p* = 0.593, NS. After NACT, the distribution of metastatic disease among those patient with a BRCA 1/2 germline mutation was comparable to those without a BRCA 1/2 germline mutation. Patients with MD, APD, and UAD were equally distributed in the group of patients with and without a BRCA 1/2 germline mutation, *p* = 0.478, NS. In just under 60% of the patients CC 0–1 was achieved and there was no difference between the groups of patients with and without a BRCA 1/2 germline mutation, *p* = 0.226, NS. The surgical effort, reflected by the surgical complexity score (SCS), to achieve this cytoreduction was comparable in both groups, *p* = 0.836, NS. Details are shown in Table 2. 

### 3.3. Overall and Progression-Free Survival

The 5-years’ OS and PFS for the total group of 168 patients was 37.8 ± 4.9% and 17.2 ± 3.5% months, respectively. Patients with a BRCA 1/2 and those without a BRCA 1/2 germline mutation had similar OS. The median OS was 43.5 months (95% CI 34.8–52.2 months) and 42.1 months (95% CI 34.5–49.7 months) for those with a BRCA 1/2 and those without a BRCA 1/2 germline mutation respectively, NS (Figure 2). The median PFS was similar in the group of patients with and the group without a BRCA 1/2 germline mutation, median PFS 19.0 months (95% CI 17.4–20.6 months) versus 19.0 months (95% CI 17.1–20.8 months respectively, NS (Figure 2). 

### 3.4. Neoadjuvant Chemotherapy

The chemotherapy regimen in all patients was platinum based. There was no difference in NACT regimen between the group patients with a BRCA 1/2 and those without a BRCA 1/2 germline mutation. Around 95% of the patients received 3–4 courses of CPT prior to cytoreductive surgery. A small portion of the patients received CP (5%) (Table 1). 

### 3.5. Multivariate Analysis of OS and PFS

The parameters PS, ΔCA125, radiological response to NACT (RECIST), preoperative extent of disease (MD, APD, and UAD), CRS, SCS, and RD affected OS with *p* < 0.10 in the univariate analysis. The parameters PS and CRS, *p* < 0.10, affected PFS in the univariate analysis. The other parameters (including BRCA status), referred to in the Patients and Methods section, did not affect OS and PFS. Therefore, these were not included in the multivariate analysis.

The multivariate analysis showed that a good PS (*p* < 0.001), CRS 2&3 (HR 7.496, 95% CI 2.523–22.27, *p* < 0.001; HR 4.069, 95% CI 1.388–11.93, *p* = 0.011), and complete cytoreduction (*p* = 0.017) were independent parameters for a better OS. Independent parameters for a better PFS were a good PS (*p* < 0.001), and CRS 2&3 (HR 3.898, 95% CI 1.873–8.112, *p* < 0.001; HR 2.000, 95% CI 1.433–5.862, *p* = 0.003). Further details are displayed in Table 3.

### 3.6. Surgical Cytoreduction

Irrespective of their BRCA status, patients with CC 0–1 had superior OS. Median OS for patients with CC 0–1 was 54.7 months versus 34.5 months (95% CI 24.7–44.3 months) and 33.5 months (95% CI 25.3–41.7 months) for CC 2 and CC 3, respectively (*p* < 0.001). The survival curves are illustrated in Figure 3.

## 4. Discussion

This study was unable to demonstrate a difference in histopathological chemotherapy response in advanced stage HGSOC patients, who underwent NACT and subsequent cytoreductive surgery, either with or without a BRCA 1/2 germline mutation. Neither were we able to demonstrate a difference between groups in terms of biochemical or radiological response to NACT. Surgical outcomes were comparable, and the presence of a BRCA 1/2 germline mutation in advanced HGSOC patients did not exert any positive influence on survival outcomes. 

Biochemical response to NACT, represented by ΔCA125, was comparable between the groups of patients with and without a BRCA 1/2 germline mutation. All the patients in our study received platinum based NACT. CA125 response is a common marker for chemotherapy response in ovarian cancer [22,23,24] although the value of CA125 response may not affect survival [25]. The validated histopathological marker CRS may serve as an alternative approach to quantify response to NACT [20]. We did not detect any difference in CRS between the group of patients with and the group without a BRCA 1/2 germline mutation. However, it has been reported that CRS may not be superior to other conventional parameters [26,27]. 

Radiological response using RECIST criteria is a more conventional parameter to assess response to NACT [17,28]. In our study, there was no difference in complete and partial responses to NACT between those patients with and without a BRCA1/2 germline mutation. Although clinical decision-making should not solely be based on RECIST criteria [29], it does not underperform when assessing response to NACT compared to other parameters [26]. 

The preoperative extent of disease, defined as MD, APD and UAD [18], may serve as a biological parameter for NACT response. We regarded the dissemination patterns, present at time of surgery after NACT, as a reflection of NACT response. Achieving CC 0–1 may depend on the dissemination pattern [30]. Nevertheless, in our study there was no difference in the dissemination patterns between patients with and without a BRCA1/2 germline mutation after NACT. We acknowledge that using this parameter for NACT response should be approached with caution since this parameter is not developed and tested in a NACT regimen for advanced HGSOC. Alternatively, the surgical effort to achieve CC 0–1 represented by SCS [20] may equally reflect the pre-operative tumor load and dissemination after NACT, and therefore serve as a parameter of biological response to NACT. Again, there were no differences in SCS between the groups of patients in our study. An equal percentage of patients with CC 0–1 in both groups (just under 60%), as confirmed by others [31] might be in support of an equal biological response to NACT comparing the groups of patients with and the group without a BRCA1/2 germline mutation. However, we do concede that response to NACT may not be fully captured by these parameters. 

Our study could not provide evidence for improved histopathological chemotherapy response amongst patients with BRCA1/2 germline mutation in comparison to those without. None of the potential indicative parameters showed an improved response to NACT in patients with a BRCA 1/2 germline mutation. These findings are partly supported by previous observations that solely HGSOC patients with a BRCA 2 germline mutation show increased platinum sensitivity, whereas patients with a BRCA 1 germline mutation do not [32]. It may well be that the number of patients with a BRCA 2 germline mutation was underrepresented in our study. 

Nevertheless, our study challenges the previous report that patients with a BRCA1/2 germline mutation have a better response to first-line chemotherapy compared to the untested patients [6]. This might be partly explained by the heterogeneity of their study population. The relationship between HGSOC and BRCA 1/2 germline mutations has been well documented, whereas this relationship with low grade, mucinous, endometrioid, and clear-cell EOC is less apparent. In contrast to our study, the aforementioned study [6] was not stratified and included patients with diverse histopathological features. Not testing for BRCA 1/2 in their EOC patients may further explain possible disparities between that study and ours. 

The OS in the groups of patients with and without a BRCA 1/2 germline mutation was similar with a median OS of 43 months. Likewise, the PFS was also comparable with a median PFS of 19 months. These observations were supported by previous studies [33,34]. In contrast, others reported improved survival in patients with a BRCA1/2 germline mutation [6,35]. However, in these studies the aim of surgical cytoreduction was frequently < 1 cm RD, whilst CC 0 was the aim in our study. Although upfront surgical cytoreduction may offer the best prognosis in advanced EOC [36], our study focused on patients receiving NACT for the purpose of assessing the histopathological CRS. Our observations may suggest that maximal effort surgical cytoreduction may eliminate the potential negative influence of an absent BRCA 1/2 germline mutations on survival. 

The limited influence of a BRCA1/2 germline mutation on OS and PFS in our population was further supported by our multivariate analysis. A BRCA1/2 germline mutation proved not to be prognostic for OS and PFS in our cohort of advanced HGSOC patients. The covariates affecting OS were CC 0–1, CRS, and PS. Surgeons may have a role in OS by maximizing complete cytoreduction rates. A more aggressive approach with maximal surgical effort may offer better outcomes in patients with advanced HGSOC, irrespective of their BRCA germline status. The median OS in our study was 43 months which is comparable to the group receiving NACT in the SCORPION trial [37] and to our previous published cohort [38]. We report a median OS of 55 months for advanced HGSOC patients with a complete cytoreduction. Although our treatment outcomes in HGSOC are satisfactory, further improvements in outcome are yet needed. Increasing the complete surgical cytoreduction rates in HGSOC patients by using conventional [39] and machine-learning based models [40], enabling the stratification patients to either upfront cytoreduction or NACT strategies as well as personalized treatment, may further improve survival rates.

The stronghold of our comparative analysis is that all included advanced stage HGSOC patients received NACT which equally served as a vehicle to assess CRS in those with and without a BRCA1/2 germline mutation. The downside of stratification in our study is the modest number of patients that fulfilled inclusion and exclusion criteria. We acknowledge that the number of the BRCA mutated patients was small (23.2%, yet within the known incidence of BRCA mutation in such cohort of patients). For that same reason, there was no further attempt to compare the survivals between BRCA1 and BRCA 2 patients. We cannot rule out that the reported lack of differences in chemotherapy response and survival between the groups of patients with and without a BRCA1/2 germline mutation may be due to an expected difference in age. In general, younger HGSOC patients do have the most favorable survival outcomes whereas the group of patients with a BRCA1/2 germline mutation in our study appeared to be on average younger. Performance status may be a more preferable parameter since despite the difference in age, the similar PS of patients in both groups of patients may explain the comparable outcome in both groups. 

The group studied included only patients receiving NACT, who then went on to cytoreductive surgery, enabling the use of histological chemotherapy response score, as an outcome measure. Of course this has selected for the group of patients with a sufficient response to chemotherapy to be eligible for surgery. It is reflected by the high radiological response rate, and very low rates of disease progression. Routine germline BRCA gene testing at the time of diagnosis was introduced at our center in 2016. Patients diagnosed before this date would have undergone testing at a later stage in their disease, sometimes to determine their eligibility for treatment with a PARP inhibitor, for which response to subsequent platinum based chemotherapy is a prerequisite. The selection of patients with intrinsically chemo sensitive disease may be reflected by the high germline BRCA mutation rate of 23%. Routine somatic BRCA gene testing and testing for homologous recombination repair (HRD) was not routine during this study. In accordance with other studies, we therefore propose to consider BRCA mutation status with regards to patient selection for cytoreductive surgery [41]. The value of this approach in the recurrent HGSOC remains to be determined. The relatively high radiological partial response rate to NACT of both groups of patients in our study may suggest the presence of somatic BRCA mutations or HRD in the group of patients without a BRCA 1/2 germline mutation [42]. Lastly, as we do not offer HIPEC at our center, it is difficult to speculate as to which of these patients would have benefited from HIPEC.

## 5. Conclusions

Our study was unable to demonstrate a difference in chemotherapy response between those patients with and those without a BRCA 1/2 germline mutation following NACT in advanced stage HGSOC patients. This is to our knowledge one of the very few studies that compared the histopathological chemotherapy response of patients with and without a BRCA 1/2 germline mutation by using NACT in advanced HGSOC. No differences in any of the other assessed variables after NACT were observed. Even so, OS in patients with and without a BRCA 1/2 germline mutation was comparable. A complete surgical cytoreduction offers patients superior survival outcomes in HGSOC, irrespective of their BRCA status. Stratification to specific geno- and phenotypes of EOC should be addressed in future surgical studies.

## Figures and Tables

**Figure 1 medicina-58-01611-f001:**
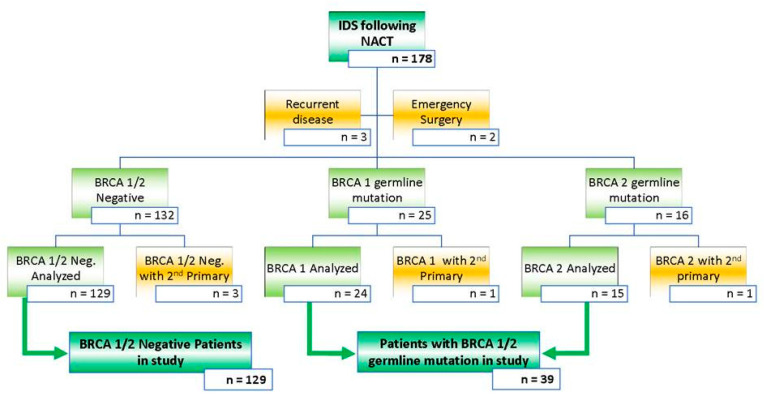
Inclusion and exclusion criteria for all patients with advanced stage HGSOC who had NACT followed by cytoreductive surgery between October 2013 and October 2018. Exclusion criteria were applied aiming at a study population of patients who had cytoreductive surgery either with or without a BRCA 1/2 germline mutation. This is a figure. Schemes follow the same formatting.

**Figure 2 medicina-58-01611-f002:**
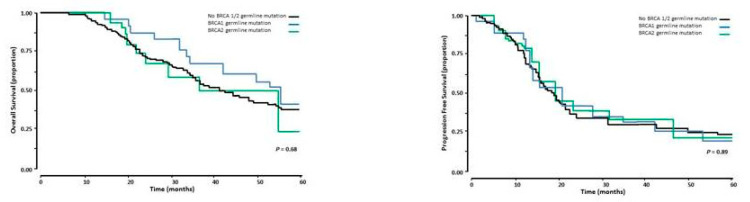
Overall survival (**left panel**) and Progression free survival (**right panel**) in 168 patients with advanced stage HGSOC who had cytireductive surgery following NACT. The black line represents the patients without a BRCA ½ mutation, the blue line those with a BRCA 1 germline mutation, and the green line those with a BRCA 2 germline mutation.

**Figure 3 medicina-58-01611-f003:**
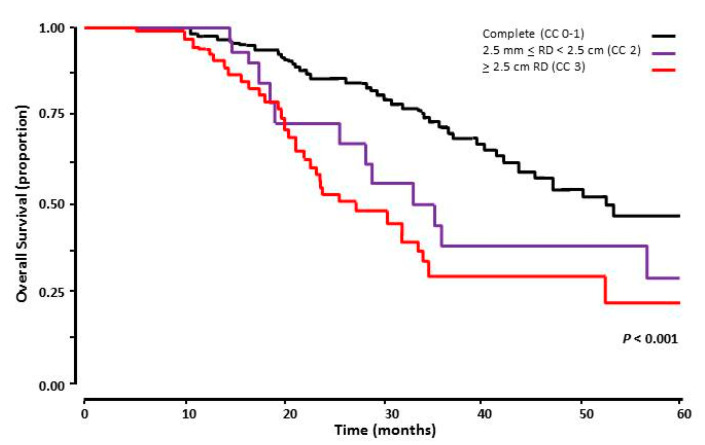
Overall survival of 168 patients with advanced stage HGSOC who had cytoreductive surgery following NACT, comparing complete cytoreduction to macroscopic residual disease (RD). The black line represents of patients with complete surgical cytoreduction following surgical cytoreduction after NACT (CC 0–1), the purple line those with 2.5 mm < RD < 2.5 cm (CC 2), and the red line the ones with ≥ 2.5 cm RD (CC 3).

**Table 1 medicina-58-01611-t001:** Baseline characteristics of 168 advanced stage (FIGO III/IV) high-grade serous ovarian cancer (HGSOC) patients; 39 patients with a BRCA ½ germline mutation and 129 patients without.

	BRCA 1/2 Mutation		No BRCA 1/2 Mutation		Total	*p*-Value
Patients	*n* = 39	23.2%	*n* = 129	76.8%	*n* = 168	
Age (yrs)	58.0 ± 7.79		66.3 ± 8.79		64.4 ± 9.79	0.00001
WHO PS						0.815
PS 0	18	46.2%	57	44.2%	75	
PS 1	13	33.3%	47	36.4%	60	
PS 2	7	17.9%	18	14.0%	25	
PS 3	1	2.6%	7	5.4%	8	
CA125 (U/mL) *	1245 (96–31,600)		791 (23–17,900)		1019 (23–31,600)	0.247
Primary Tumor						0.722
Ovary	21	53.8%	60	46.5%	81	
Peritoneum	14	35.9%	53	41.1%	67	
Fallopian Tube	4	10.3%	16	12.4%	20	
FIGO Stage						0.584
III A-B	1	2.6%	5	3.9%	6	
III C	24	61.5%	82	63.6%	106	
IV A	7	17.9%	12	9.3%	19	
IV B	7	17.9%	30	23.3%	37	
Tumor Differentiation						1.000
Well	0	0%	0	0%	0	
Moderate	0	0%	1	0.8%	1	
Poor	39	100%	128	99.2%	167	
Agent NACT						0.829
CP	2	5.1%	7	5.4%	9	
CPT	37	94.9%	121	94.6%	158	

Except for *, numbers are shown either as absolute numbers with percentage or as mean with standard deviation. * CA125 numbers are shown as median (min-max).

**Table 2 medicina-58-01611-t002:** Biochemical, radiological, histopathological, biological, and surgical parameters of platinum sensitivity in 168 advanced stage HGSOC patients following neoadjuvant chemotherapy (NACT); 39 patients with a BRCA ½ germline mutation and 129 without.

	BRCA 1/2 Mutation		No BRCA 1/2 Mutation		Total	*p*-Value
Patients	*n* = 39	23.2%	*n* = 129	76.8%	*n* = 168	
ΔCA125 (U/mL) *	1111 (27–30715)		641 (2–15897)		801 (2–30,715)	0.981
RECIST after NACT						0.677
Complete	0	0.0%	3	2.3%	3	
Partial	35	89.7%	108	83.7%	143	
Stable disease	3	7.7%	16	12.4%	19	
Prog. Disease	1	2.6%	2	1.6%	3	
Preoperative Extent of Disease						0.478
MD	1	2.6%	8	6.2%	9	
APD	33	84.6%	98	76.0%	131	
UAD	5	12.8%	23	12.4%	28	
Histological response						0.593
CRS 1	17	43.6%	48	37.2%	65	
CRS 2	18	46.1%	60	46.5%	78	
CRS 3	4	10.3%	21	16.3%	25	
SCS Group						0.836
Low	29	74.4%	98	76.0%	127	
Intermediate	8	20.5%	27	20.9%	35	
High	2	5.1%	4	3.1%	6	
Residual Disease (RD)						0.226
0–2.5 mm (CC 0–1)	22	56.4%	76	58.9%	98	
≥2.5 mm (CC 2)	7	18.0%	11	8.5%	18	
≥2.5 cm (CC 3)	10	25.6%	42	32.6%	52	

Except for *, numbers are shown either as absolute numbers with percentage or as mean with standard deviation. * CA125 numbers are shown as median (min-max).

**Table 3 medicina-58-01611-t003:** Multivariate analysis of covariates for OS and PFS. Covariates from all available parameters (including BRCA ½ status) were selected when *p* < 0.1 was reached in the univariate analysis for OS.

	Multivariate Analysis OS			Multivariate Analysis PFS		
Covariates	HR	*p*	95% CI	HR	*p*	95% CI
Performance Score						
PS 0	1.000	<0.001		1.00	<0.001	
PS 1	0.134	<0.001	0.051–0.353	0.205	<0.001	0.082–0.514
PS 2	0.490	0.019	0.209–0.929	0.218	0.004	0.077–0.618
Result of Cytoreduction						
Completete (CC 0–1)	1.000	0.017		1.00	0.540	
≥2.5 mm RD (CC 2)	0.446	0.175	0.139–1.430	0.710	0.537	0.239–2.107
≥2.5 cm RD (CC 3)	0.701	0.428	0.292–1.685	0.853	1.087	0.449–2.636
Chemotherapy Response Score						
CRS 1	1.000	<0.001		1.00	<0.001	
CRS 2	7.496	<0.001	2.523–22.27	3.898	<0.001	1.873–8.112
CRS 3	4.069	0.011	1.388–11.93	2.000	0.003	1.433–5.862
Preoperative Extent of Disease						
MD	1.00	0.17		1.00	0.105	
APD	0.235	0.082	0.046–1.201	0.902	0.845	0.320–2.542
UAD	1.750	0.132	0.846–3.622	1.713	0.084	0.931–3.150
Radiological Response						
Complete (RECIST)	1.00	0.762		1.00	0.487	
Partial/Stable (RECIST)	4.271	0.254	0.352–51.86	0.503	0.533	0.058–4.367
Progressive (RECIST)	0.176	0.124	0.019–1.608	0.444	0.197	0.129–1.526
Surgical Complexity Score						
Low Complexity	1.00	0.542		1.00	0.893	
Intermediate Complexity	0.886	0.847	0.258–3.038	1.240	0.646	0.494–3.112
High Complexity	1.240	0.746	0.338–4.544	1.191	0.727	0.447–3.173
ΔCA125 (U/mL)	1.107	0.695	0.666–1.839	1.179	0.404	0.800–1.739

## Data Availability

Data are available from the corresponding author upon reasonable request.

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
