# Peer review of "Survival and Chemosensitivity in Advanced High Grade Serous Epithelial Ovarian Cancer Patients with and without a BRCA Germline Mutation: More Evidence for Shifting the Paradigm towards Complete Surgical Cytoreduction"

_medicina, 2022, doi:10.3390/medicina58111611_

Round 1

Reviewer 1 Report

I'd like to commend the authors on their well-written paper.

The methodology is described clearly, the results are well-presented, and the conclusion with limitations is well-written and balanced.

I have no additional comments, and I believe the paper should be published in its current form.

Author Response

Many thanks for the positive feedback from the reviewer.

Reviewer 2 Report

This is an article that compared the prognosis of patients with at least stage III ovarian cancer according to germline BRCA mutation status.

Why did you not include somatic BRCA-mutated patients?

In your population, are there any patients who have benefited from HIPEC?

The number of mutated patients is relatively small and to separate survival analysis between BRCA1 and BRCA2 has low statistical power.

Author Response

Thank you for your valuable comments. 

The setting for testing for somatic BRCA mutations was not available at the time.

As we do not yet offer HIPEC at our center, it is difficult to say which of these patients would have benefited from HIPEC.

We appreciate the number of the BRCA mutated patients was small (39/162, yet 23.2% within the known incidence of BRCA mutation in a such cohort of patients. For that same reason, there was no further attempt to check survivals between BRCA1 and BRCA 2 patients. Please see the attachment
